# MINI-BATCH SUBMODULAR MAXIMIZATION

## ABSTRACT

We present the first *mini-batch* algorithm for maximizing a non-negative monotone *decomposable* submodular function, $F = \sum_{i=1}^{N} f^i$, under a set of constraints. We consider two sampling approaches: uniform and weighted. We first show that mini-batch with weighted sampling improves over the state of the art sparsifier based approach both in theory and in practice. Surprisingly, our experimental results show that uniform sampling is superior to weighted sampling. However, it is *impossible* to explain this using worst-case analysis. Our main contribution is using *smoothed analysis* to provide a theoretical foundation for our experimental results. We show that, under *very mild* assumptions, uniform sampling is superior for both the mini-batch and the sparsifier approaches. We empirically verify that these assumptions hold for our datasets. Uniform sampling is simple to implement and has complexity independent of $N$, making it the perfect candidate to tackle massive real-world datasets.

## 1 INTRODUCTION

Submodular functions capture the natural property of *diminishing returns* which often arises in machine learning, graph theory and economics. For example, imagine you are given a large set of images, and your goal is to extract a small subset of images, that best represent the original set (e.g., creating thumbnails for a Youtube video). Intuitively, this is a submodular optimization problem, as the more thumbnails we have, the less we gain by adding an additional thumbnail.

Formally, a set function $F : 2^E \to \mathbb{R}^+$, on a *ground set* $E$, is submodular if for any subsets $S \subseteq T \subseteq E$ and $e \in E \setminus T$, it holds that $F(S + e) - F(S) \geq F(T + e) - F(T)$.

**Decomposable submodular functions** In many natural scenarios $F$ is *decomposable*: $F = \sum_{i=1}^{N} f^i$, where each $f^i : 2^E \to \mathbb{R}^+$ is a non-negative submodular function on the ground set $E$ with $|E| = n$. That is, $F$ can be written as a sum of "simple" submodular functions. We assume that every $f^i$ is represented by an evaluation oracle that when queried with $S \subseteq E$ returns the value $f^i(S)$. For ease of notation we assume that $f^i(\emptyset) = 0$ (our results hold even if this not the case). Our goal is to maximize $F$ under some set of constraints while minimizing the number of oracle calls to $\{f^i\}$. An excellent survey of the importance of decomposable functions is given in (Rafiey & Yoshida, 2022), which we summarize below.

Decomposable submodular functions are prevalent in both machine learning and economics. In economics, they play a pivotal role in welfare optimization during combinatorial auctions (Dobzinski & Schapira, 2006; Feige, 2009; Feige & Vondrák, 2006; Papadimitriou et al., 2008; Vondrák, 2008). In machine learning, these functions are instrumental in tasks like data summarization, aiming to select a concise yet representative subset of elements. Their utility spans various domains, from exemplar-based clustering by (Dueck & Frey, 2007) to image summarization (Tschiatschek et al., 2014), recommender systems (Parambath et al., 2016) and document summarization (Lin & Bilmes, 2011). The optimization of these functions, especially under specific constraints (e.g., cardinality, matroid) has been studied in various data summarization settings (Mirzasoleiman et al., 2016a;b;c) and differential privacy (Chaturvedi et al., 2021; Mitrovic et al., 2017; Rafiey & Yoshida, 2020). In many of the above applications $N$ (the number of underlying submodular functions) is extremely large, making the evaluation of $F$ prohibitively slow. We illustrate this with a simple example.

**Example (Welfare maximization)** Imagine you are tasked with deciding on a meal menu for a large group of $N$ people (e.g., all students in a university, all high school students in a country). You need to choose $k$ ingredients to use from a predetermined set (chicken, fish, beef, etc.) of size $n$.

Every student has a specific preference, modeled as a monotone submodular function $f^i$. Our goal is to maximize the "social welfare", $F(S) = \sum f^i(S)$, over all students.

**The greedy algorithm**     Going forward we focus on the case where $\forall i, f^i$ is monotone ($\forall S \subseteq T \subseteq E, f^i(T) \geq f^i(S)$), which in turn means $F$ is monotone. This is applicable to the above example (e.g., students are happier with more food varieties). For ease of presentation let us first focus on maximizing $F$ under a cardinality constraint $k$, i.e., $\max F(S), |S| \leq k$. The classical greedy algorithm (Nemhauser et al., 1978) achieves an *optimal* $(1 - 1/e)$-approximation for this problem. For $S, A \subseteq E$ we define $F_S(A) = F(S + A) - F(S)$. We slightly abuse notation and write $F_S(e), F(e)$ instead of $F_S(\{e\}), F(\{e\})$.

**Greedy:**     Start by initializing an empty set $S_1$. Then for each $j$ from 1 to $k$, perform the following steps: First, identify the element $e'$ not already in $S_j$ that maximizes the function $F_{S_j}(e)$. Add this element $e'$ to set $S_j$ to form the new set $S_{j+1}$. After completing all iterations, return the set $S_{k+1}$.

When $F$ is decomposable, and each evaluation of $f^i$ is counted as an oracle call, the above algorithm requires $O(Nnk)$ oracle calls. This can be prohibitively expensive if $N \gg n$. Looking back at our example, Greedy will require asking every student $k$ questions. That is, we would start by asking all students: "Rate how much you would like to see food $x$ on the menu" for $n$ possible options. We would then need to wait for all of their replies, and continue with "Given that chicken is on the menu, rate how much you would like to see food $y$ on the menu" for $n - 1$ possible options and so on. Even if we do an online poll, this is still very time-consuming (as we must wait for *everyone* to reply in each of the $k$ steps). If we could *sample* a representative subset of the students, we could greatly speed up the above process. Specifically, we would like to eliminate the dependence on $N$.

**A sparsifier based approach**     Recently, Rafiey & Yoshida (2022) were the first to consider constructing a *sparsifier* for $F$. That is, given a parameter $\epsilon > 0$ they show how to find a vector $w \in \mathbb{R}^N$ such that the number of non-zero elements in $w$ is small in expectation and the function $\hat{F} = \sum_{i=1}^{N} w_i f^i$ satisfies with high probability (w.h.p)[1] that $\forall S \subseteq E, (1 - \epsilon)F(S) \leq \hat{F}(S) \leq (1+\epsilon)F(S)$. Where the main bottleneck in the above approach is computing the sampling probabilities for computing $w$. They treat this step as a *preprocessing* step. The current state of the art construction is due to (Kenneth & Krauthgamer, 2023) where a sparsifier of size $O(k^2 n \epsilon^{-2})$ (where $k$ bounds the size of the solution) can be constructed using $O(Nn)$ oracle calls (see Appendix A for an in-depth overview of existing sparsifier constructions).

**Weighted mini-batch**     Mini-batch methods are at the heart of several fundamental machine learning algorithms (e.g., mini-batch k-means, SGD). Surprisingly, a mini-batch approach for this problem was not considered. We present the first mini-batch algorithm for this problem, and show that it is superior to the sparsifier based approach both in theory and in practice. Roughly speaking, we show that sampling a new batch every iteration of the algorithm is "more stable" compared to the sparsifier approach. This allows us to sample less elements in total and reduce the overall complexity. The main novelty in our analysis is using the greedy nature of the algorithm and carefully balancing an additive and a multiplicative error term. Our sampling probabilities are the same as those of (Kenneth & Krauthgamer, 2023), and therefore, this algorithm pays the expensive $O(Nn)$ preprocessing time.

**Beyond worst-case analysis**     When conducting our experiments, we added a simple baseline for both the sparsifier and the mini-batch algorithm – instead of using weighted sampling we used *uniform sampling* (previous results neglected this baseline (Rafiey & Yoshida, 2022)). Surprisingly, we observe that it outperforms weighted sampling both for the sparsifier and mini-batch algorithms. This is remarkable, as uniform sampling is extremely simple to implement, and requires no preprocessing which removes the dependence on $N$ altogether.

Unfortunately, we cannot get worst-case theoretical guarantees for uniform sampling. Consider the case where only a single $f^j$ takes non-zero values (all other $f^i$'s are always 0). Clearly, uniform sampling will almost surely miss $f^j$. However, this is a pathological case that is very unlikely to occur in practice. To bridge this gap we go beyond worst-case analysis and consider the *smoothed complexity* of this problem.

Smoothed analysis was introduced by (Spielman & Teng, 2004) in an attempt to explain the fast runtime of the Simplex algorithm in practice, despite its exponential worst-case runtime. In the

---

[1] Probability at least $1 - 1/n^c$ for an arbitrary constant $c > 1$. The value of $c$ does affect the asymptotics of the results we state (including our own).

smoothed analysis framework, an algorithm is provided with an adversarial input that is perturbed by some random noise. The crux of smoothed analysis is often defining a realistic model of noise.

Our main contribution is defining two very natural smoothing models. The reason we define two models is because the first allows us to provide theoretical guarantees both for our mini-batch algorithms and for all existing sparsifier algorithms (under uniform sampling), but only lines up empirically with some of the datasets we use. For the second model we are only able to show theoretical guarantees for our mini-batch algorithm, but it agrees empirically with all of our datasets.

To define our models of smoothing we assume w.log that $\forall i \in [N], e \in E, f^i(e) \in [0, 1]$ (we can always achieve this by normalization if $\{f^i(e)\}$ are upper bounded). Let $\phi \in [0, 1], d \in [N]$ be parameters and let us denote $A_e = \{f^i(e)\}_{i \in [N]}$.

**Model 1**    It holds that $N = \Omega(\frac{d}{\phi} \log(nd))$, and for *every* $e \in E$ the following two conditions hold: (1) Every $f^i(e) \in A_e$ is a random variable such that $\mathbb{E}[f^i(e)] \geq \phi$. (2) Elements in $A_e$ have dependency at most $d$ (every $f^i(e)$ depends on at most $d$ other elements in $A_e$). Note that we can have arbitrary dependencies between elements in $A_e, A_{e'}, e \neq e'$.

**Model 2**    Identical to Model 1, except that there *exists* $e \in E$ such that conditions (1) and (2) hold.

**Intuition**    Going back to our lunch menu example, Assumption (1) of Model 1 means that every possible food on the menu is not universally hated by the students. Assumption (2) means that the preference of a student for any specific food is sufficiently independent of the preferences of other students. Assuming that $N = \Omega(\frac{d}{\phi} \log(nd))$ means that we have a sufficiently large student body – note that increasing $N$ should not change $\phi$ and $d$. Model 2 only requires assumptions (1) and (2) to hold for a single menu item. Intuitively, Model 1 assumes that all food choices are "not too bad" while Model 2 assumes that there exists at least one such choice.

**Comparison to other model of smoothing**    Perhaps the most general smoothing approach when dealing with weighted inputs (e.g., in [0,1]) is to assume that the weights are taken from some distribution whose density is upper bounded by a smoothing parameter $\phi$ (Etscheid & Röglin, 2017; Angel et al., 2017). This generalizes the approach of Spielman & Teng (2004), where Guassian noise was added to the weights. Our approach is even more general, as the above immediately implies that the expectation is lower bounded by $\phi$. Furthermore, we only assume bounded independence and Model 2 only partially smoothes the input.

While our primary contribution lies in providing theoretical guarantees for the uniform mini-batch algorithm (which empirically outperforms all other methods), we begin by presenting our results for weighted sampling. This will lay the groundwork to seamlessly prove our main results in Section 4.

## 1.1 THE MINI-BATCH ALGORITHM

We focus on the *greedy algorithm* for constrained submodular maximization. We show that instead of sparsifying $F$, better results can be achieved by using mini-batches during the execution of the greedy algorithm. That is, rather than sampling a large sparsifier $\hat{F}$ and performing the optimization process, we show that if we sample a much smaller sparsifier (a *mini-batch*), $\hat{F}^j$, for the $j$-th step of the greedy algorithm, we can achieve improved results both in theory (Table 1) and in practice (Section 3). Most notably, we observe that the mini-batch approach is superior to the sparsifier approach for small batch sizes on various real world datasets. This is also the case when we combine our approach with the popular *stochastic-greedy* algorithm (Buchbinder et al., 2015; Mirzasoleiman et al., 2015).

While the mini-batch approach results in an improvement in performance, the sparsifier approach has the benefit of being independent of the algorithm. That is, while any approximation algorithm executed on a sparsifier immediately achieves (nearly) the same guarantees for the original function, we need to re-establish the approximation ratio of our mini-batch algorithm for different constraints. Although these proofs are often straightforward, compiling an exhaustive list of where the mini-batch method is applicable is both laborious and offers limited insights.

We focus on two widely researched constraints: the cardinality constraint and the $p$-system constraint. The cardinality constraint was chosen for its simplicity, while the $p$-system constraint was chosen for its broad applicability.

**Theoretical results**    We compare our results with the state of the art sparsifier results and the naive algorithm (without sampling or sparsification) in Table 1[2]. We can get improved performance if the curvature of $F$ is bounded[3]. Note that the "Uniform" column requires no preprocessing, and the query complexity differs by a multiplicative $\Theta(1/n\phi)$-factor. In Section 2 we prove our results for the "Weighted" column and in Section 4 we prove our results for the "Uniform" column. Under Model 2 our results for the unbounded curvature mini-batch case still hold for uniform sampling (all other results hold under Model 1). We empirically observe that $\phi = O(1)$ for our datasets (Section 4), which explains the superior performance of the uniform mini-batch algorithm in practice.

| | Preprocessing | | Oracle queries | |
|---|---|---|---|---|
| | Weighted | **Uniform** | Weighted | **Uniform** (Model 1) |
| Naive | None | | $O(Nnk)$ | |
| Kenneth & Krauthgamer | $O(Nn)$ | None | $\widetilde{O}\left(\frac{k^3 n^2}{\epsilon^2}\right)$ | $\widetilde{O}\left(\frac{k^3 n}{\epsilon^2 \phi}\right)$ |
| **Our results** Uniform holds for Model 1 & 2 | $O(Nn)$ | None | Card. $\widetilde{O}\left(\frac{k^2 n^2}{\epsilon^2}\right)$ $p$-sys. $\widetilde{O}\left(\frac{k^2 p n^2}{\epsilon^2}\right)$ | Card. $\widetilde{O}\left(\frac{k^2 n}{\epsilon^2 \phi}\right)$ $p$-sys. $\widetilde{O}\left(\frac{k^2 p n}{\epsilon^2 \phi}\right)$ |
| **Our results** (bounded curvature) | $O(Nn)$ | None | $\widetilde{O}\left(\frac{kn^2}{(1-c)\epsilon^2}\right)$ | $\widetilde{O}\left(\frac{kn}{(1-c)\epsilon^2 \phi}\right)$ |

Table 1: Comparison of the number of oracle queries during preprocessing and during execution. Results are for the greedy algorithm under both a cardinality constraint and a $p$-system constraint. Unless explicitly stated the number of queries is the same for both constraints. All results achieve the near optimal approximation guarantees of $(1 - 1/e - \epsilon)$ for a cardinality constraint and $\left(\frac{1-\epsilon}{p+1}\right)$ for a $p$-system constraint.

**Meta greedy algorithm**    Our starting point is the meta greedy algorithm (Algorithm 1). The algorithm executes for $k \leq n$ iterations where $k$ is some upper bound on the size of the solution. At every iteration, the set $A_j \subseteq E \setminus S_j$ represents some constraint that limits the choice of potential elements to extend $S_j$. The algorithm terminates either when the solution size reaches $k$ or when no further extensions to the current solution are possible (i.e., $A_j = \emptyset$). Furthermore, the algorithm does not have access to the exact *incremental oracle*, $F_{S_j}$, at every iteration, but only to some approximation (which may differ between iterations). Before we formally define "approximation",

---

**Algorithm 1:** Meta greedy algorithm with an approximate oracle

1   $S_1 \leftarrow \emptyset$
2   Let $k$ be an upper bound on the size of the solution
3   **for** $j = 1$ *to* $k$ **do**
4      Let $A_j \subseteq E \setminus S_j$   ▷ Problem specific constraint (e.g., $A_j = E \setminus S_j$ for card. constraint)
5      **if** $A_j = \emptyset$ **then** return $S_j$
6      Let $\hat{F}_{S_j}^j$ be an approximation for $F_{S_j}$   ▷ Problem specific approximation
7      $e_j = \arg\max_{e \in A_j} \hat{F}_{S_j}^j(e)$
8      $S_{j+1} = S_j + e_j$
9   **end**
10   return $S_{k+1}$

---

let us note that when we have access to exact values of $F_{S_j}$, Algorithm 1 captures many variants of the greedy submodular maximization algorithm. For example, setting $A_j = E \setminus S_j$ we get `Greedy`. This meta-algorithm also captures the case of maximization under a *$p$-system constraint*. For ease of presentation we defer the discussion about $p$-systems to Appendix B.

---

[2]Where $\widetilde{O}$ hides $\log n$ factors.

[3]The *curvature* of a submodular function $F$ is defined as $c = 1 - \min_{S \subseteq E, e \in E \setminus S} \frac{F_S(e)}{F(e)}$. We say that $F$ has *bounded-curvature* if $c < 1$.

**Approximate oracles**    In many scenarios we do not have access to *exact* values of $F_{S_j}$, and instead we must make do with an approximation. We start with the notion of an *approximate incremental oracle* introduced in (Goundan & Schulz, 2007). We say that $\hat{F}_{S_j}^j$ is an $(1 - \epsilon)$-approximate incremental oracle if $\forall e \in A_j, (1 - \epsilon)F_{S_j}(e) \leq \hat{F}_{S_j}^j(e) \leq (1 + \epsilon)F_{S_j}(e)$. It was shown in (Goundan & Schulz, 2007; Călinescu et al., 2011)[4] that given a $(1 - \epsilon)$-approximate incremental oracle, the greedy algorithm under both a cardinality constraint and a $p$-system constraint achieves almost the same (optimal) approximation ratio as the non-approximate case.

**Theorem 1.1.** *Algorithm 1 with an $(1 - \epsilon)$-approximate incremental oracle has the following guarantees w.h.p: (1) It achieves a $(1 - 1/e - \epsilon)$-approximation under a cardinality constraint $k$ (Goundan & Schulz, 2007). (2) It achieves a $(\frac{1-\epsilon}{1+p})$-approximation under a $p$-system constraint (Călinescu et al., 2011).*

We introduce a weaker type of approximate incremental oracle, which we call an *additive* approximate incremental oracle. We extend the results of Theorem 1.1 for this case. Let $S^*$ be some optimal solution for $F$ (under the relevant set of constraints). We say that $\hat{F}_{S_j}^j$ is an *additive* $\epsilon'$-approximate incremental oracle if $\forall e \in A_j, F_{S_j}(e) - \epsilon'F(S^*) \leq \hat{F}_{S_j}^j(e) \leq F_{S_j}(e) + \epsilon'F(S^*)$.

This might seem problematic at first glance, as it might be the case that $F(S^*) \gg F_{S_j}(e)$. Luckily, the proofs guaranteeing the approximation ratio are *linear* in nature. Therefore, by the end of the proof we end up with an expression of the form: $F(S_{k+1}) \geq F(S^*)\beta - \gamma\epsilon'F(S^*)$. Where $\beta$ is the desired approximation ratio and $\gamma$ depends on the parameters of the problem (e.g., $\beta = (1 - 1/e), \gamma = 2k$ for a cardinality constraint). We can achieve the desired result by setting $\epsilon' = \epsilon/\gamma$. We state the following theorem (the proofs are similar to those of (Goundan & Schulz, 2007; Călinescu et al., 2011), and we defer them to the Appendix).

**Theorem 1.2.** *Algorithm 1 with an additive $\epsilon'$-approximate incremental oracle has the following guarantees w.h.p: (1) If $\epsilon' < \epsilon/2k$, it achieves a $(1 - 1/e - \epsilon)$-approximation under a cardinality constraint $k$. (2) If $\epsilon' < \epsilon/2kp$, it achieves a $(\frac{1-\epsilon}{1+p})$-approximation under a $p$-system constraint.*

**Mini-batch sampling**    Although we use the same sampling probabilities as the sparsifier approach, instead of sampling a single $\hat{F}$ at the beginning, we sample a new $\hat{F}^j$ (*mini-batch*), for every step of the algorithm. Recall that $\hat{F}_{S_j}^j(e) = \hat{F}^j(S_j + e) - \hat{F}^j(S_j)$.

We show that when $\hat{F}_{S_j}^j$ is sampled using mini-batch sampling we indeed get, w.h.p, an (additive) approximate incremental oracle for every step of the algorithm. We present our sampling procedure in Algorithm 2 and the complete mini-batch algorithm in Algorithm 3.

---

**Algorithm 2:** Sample($\alpha, \{p_i\}_{i=1}^N$)

1  $w \leftarrow \vec{0}$
2  **for** $i = 1$ *to* $N$ **do**
3  $\quad$ $\alpha_i \leftarrow \min\{1, \alpha p_i\}$
4  $\quad$ $w_i \leftarrow 1/\alpha_i$ with probability $\alpha_i$
5  **end**
6  return $\hat{F} = \sum_{i=1}^N w_i f^i$

---

**Algorithm 3:** Mini-batch greedy

1  $\forall i \in [N], p_i \leftarrow \max_{e \in E, F(e) \neq 0} \frac{f^i(e)}{F(e)}$
$\quad$ // Uniform sampling:  $p_i = 1/N$
2  $\alpha$ is a batch parameter
3  $S_1 \leftarrow \emptyset$
4  $k$ is an upper bound on the size of the solution
5  **for** $j = 1$ *to* $k$ **do**
6  $\quad$ Let $A_j \subseteq E \setminus S_j$
7  $\quad$ **if** $A_j = \emptyset$ **then** return $S_j$
8  $\quad$ Let $\hat{F}^j \leftarrow Sample(\alpha, \{p_i\}_{i=1}^N)$
9  $\quad$ $e_j = \arg\max_{e \in A_j} \hat{F}_{S_j}^j(e)$
10  $\quad$ $S_{j+1} = S_j + e_j$
11  **end**
12  return $S_{k+1}$

---

[4]Strictly speaking, both Goundan & Schulz (2007) and Călinescu et al. (2011) define the approximate incremental oracle to be a function that returns $e_j$ at iteration $j$ of the greedy algorithm such that $\forall e \in A_j, F_{S_j}(e_j) \geq (1 - \epsilon)F_{S_j}(e)$. Our definition guarantees this property while allowing easy analysis of the mini-batch algorithm.

In Section 2 we analyze the relation between the batch parameter, $\alpha$, and the the type of approximate incremental oracles guaranteed by our sampling procedure. We state the main theorem for the section below.

**Theorem 1.3.** *The mini-batch greedy algorithm (Algorithm 3) maximizing a non-negative monotone submodular function has the following guarantees:*

1. *If $F$ has curvature bounded by $c$, and $\alpha = \Theta(\frac{\log n}{\epsilon^2(1-c)})$ it holds w.h.p that $\forall j \in [k]$ that $\hat{F}_{S_j}^j$ is an $(1-\epsilon)$-approximate incremental oracle.*

2. *If $\alpha = \Theta(\epsilon^{-2}\gamma \log n)$ it holds w.h.p that $\forall j \in [k]$ that $\hat{F}_{S_j}^j$ is an additive $(\epsilon/\gamma)$-approximate incremental oracle, for any parameter $\gamma > 0$.*

*Furthermore, the number of oracle evaluations during preprocessing is $O(nN)$ and an expected $\alpha(\sum_{i=1}^N p_i)(\sum_{j=1}^k |A_j|) = O(\alpha k n^2)$ during execution.*

Combining Theorem 1.3 (setting $\gamma = k$ for a cardinality constraint and $\gamma = kp$ for a $p$-system constraint) with Theorem 1.1 and Theorem 1.2 we state our main result.

**Theorem 1.4.** *The mini-batch greedy algorithm maximizing a non-negative monotone submodular function requires $O(nN)$ oracle calls during preprocessing and has the following guarantees:*

1. *If $F$ has curvature bounded by $c$, it achieves w.h.p a $(1-1/e-\epsilon)$-approximation under a cardinality constraint and $(\frac{1-\epsilon}{1+p})$-approximation under a $p$-system constraint with an expected $O(\frac{kn^2 \log n}{\epsilon^2(1-c)})$ oracle evaluations for both cases.*

2. *It achieves w.h.p a $(1-1/e-\epsilon)$-approximation under a cardinality constraint and $(\frac{1-\epsilon}{1+p})$-approximation under a $p$-system constraint with an expected $O(k^2(n/\epsilon)^2 \log n)$ and $O(k^2 p(n/\epsilon)^2 \log n)$ oracle evaluations respectively.*

### 1.2 RELATED WORK

**Approximate oracles**  Apart from the results of (Goundan & Schulz, 2007; Călinescu et al., 2011) there are works that use different notions of an approximate oracle. Several works consider an approximate oracle $\hat{F}$, such that $\forall S \subseteq E, \left|\hat{F}(S) - F(S)\right| < \epsilon F(S)$ (Crawford et al., 2019; Horel & Singer, 2016; Qian et al., 2017). The main difference of these models to our work is the fact that they do not assume the surrogate function, $\hat{F}$, to be submodular. This adds a significant complication to the analysis and degrades the performance guarantees.

**Mini-batch methods**  The closest result resembling our mini-batch approach is the *stochastic-greedy algorithm* (Buchbinder et al., 2015; Mirzasoleiman et al., 2015). They improve the expected query complexity of the greedy algorithm under a cardinality constraint by only considering a small random subset of $E \setminus S_j$ at the $j$-th iteration. We note that their approach can be combined into our mini-batch algorithm, reducing our query complexity by a $\tilde{\Theta}(k)$ factor, resulting in an approximation guarantee in expectation instead of w.h.p.

**Smoothed analysis**  To the best of our knowledge, (Rubinstein & Zhao, 2022) is the only result that considers smoothed analysis in the context submodular maximization. They consider submodular maximization under a cardinality constraint, where the cardinality parameter $k$ undergoes a perturbation according to some known distribution.

## 2 ANALYSIS OF THE MINI-BATCH GREEDY ALGORITHM

We start by with the following lemma from (Kenneth & Krauthgamer, 2023), which bounds the expected size of $\hat{F}$. We present a proof in the appendix for completeness.

**Lemma 2.1.** *The expected size of $\hat{F}$ is $\alpha \sum_{i=1}^N p_i \leq \alpha n$.*

Note that the above yields a tighter bound of $\alpha$ for uniform sampling ($p_i = 1/N$). Next, let us show that $\hat{F}$ returned by Algorithm 2 is indeed an (additive) approximate incremental oracle w.h.p. We make use of the following Chernoff bound.

**Theorem 2.2** (Chernoff bound (Motwani & Raghavan, 1995)). *Let $X_1, ..., X_N$ be independent random variables in the range $[0, a]$. Let $X = \sum_{i=1}^{N} X_i$. Then for any $\epsilon \in [0, 1]$ and $\mu \geq \mathbb{E}[T]$ it holds that $\mathbb{P}(|X - \mathbb{E}[X]| \geq \epsilon\mu) \leq 2\exp\left(-\epsilon^2\mu/3a\right)$.*

The following lemma provides concentration guarantees for $\hat{F}$ in Algorithm 2.

**Lemma 2.3.** *For every $S \subseteq E$ ($\hat{F}$ sampled after $S$ is fixed) and for every $e \in E$ and $\mu \geq F_S(e)$, it holds that $\mathbb{P}\left[|\hat{F}_S(e) - F_S(e)| \geq \epsilon\mu\right] \leq 2\exp\left(-\frac{\epsilon^2\mu}{3F(e)/\alpha}\right)$.*

*Proof.* Fix some $e \in E$. Let $G = \sum_{i \in I} f^i$, where $I = \{i \in [N] \mid \alpha_i = 1\}$. Let $F'_S(e) = F_S(e) - G_S(e)$ and $\hat{F}'_S(e) = \hat{F}_S(e) - G_S(e)$. Let $J = [N] \setminus I$. It holds that:

$$\mathbb{P}\left[|\hat{F}_S(e) - F_S(e)| \geq \epsilon\mu\right] = \mathbb{P}\left[|\hat{F}'_S(e) + G_S(e) - F'_S(e) - G_S(e)| \geq \epsilon\mu\right] = \mathbb{P}\left[|\hat{F}'_S(e) - F'_S(e)| \geq \epsilon\mu\right]$$

Due to the fact that $\mathbb{E}[w_i] = 1$ we have $\mathbb{E}[\hat{F}'_S(e)] = \mathbb{E}[\sum_{i \in J} w_i f^i_S(e)] = F'_S(e)$. As $f^i$'s are monotone, it holds that $\mu \geq F_S(e) \geq F'_S(e)$. Applying a Chernoff bound (Theorem 2.2) we have

$$\mathbb{P}\left[|\hat{F}'_S(e) - F'_S(e)| \geq \epsilon\mu\right] \leq 2\exp\left(-\epsilon^2\mu/3a\right)$$

where $a = \max\{w_i f^i_S(e)\}_{i \in J}$. Recall that $w_i = 1/\alpha_i$ where $\alpha_i = \min\{1, \alpha p_i\}$ and $\alpha_i < 1$ for all $i \in J$. Let us upper bound $a$.

$$a = \max_{i \in J} w_i f^i_S(e) = \max_{i \in J} \frac{f^i_S(e)}{\alpha p_i} = \max_{i \in J} \frac{f^i_S(e)}{\alpha \cdot \max_{e' \in E} \frac{f^i(e')}{F(e')}} \leq \max_{i \in J} \frac{f^i(e)}{\alpha \cdot \frac{f^i(e)}{F(e)}} = \frac{F(e)}{\alpha}$$

Where the inequality is due to submodularity and non-negativity in the numerator and maximality in the denominator. Note that the above is also correct if we only have some approximation to $p_i$ – i.e., given $p'_i > p_i\lambda$ for $\lambda \in (0, 1)$, we can increase $\alpha$ by a $1/\lambda$ factor and the above still holds. Given the above upper bound for $a$ we get:

$$\mathbb{P}\left[|\hat{F}_S(e) - F_S(e)| \geq \epsilon\mu\right] \leq 2\left(-\frac{\epsilon^2\mu}{3a}\right) \leq 2\exp\left(-\frac{\epsilon^2\alpha\mu}{3F(e)}\right) \qquad \square$$

Using the above we are ready to prove Theorem 1.3.

***Proof of Theorem 1.3.*** The number of oracle evaluations is due to Lemma 2.1 and the fact that the algorithm executes for $k$ iteration and must evaluate $|A_j| \leq n$ elements per iteration.

Let us prove the approximation guarantees. Let us start with the bounded curvature case. Fix some $S_j$. As $\hat{F}^j$ is sampled after $S_j$ is fixed, we can fix some $e \in E$ and apply Lemma 2.3 with $\mu = F_{S_j}(e)$. We get that:

$$\mathbb{P}\left[|\hat{F}^j_{S_j}(e) - F_{S_j}(e)| \geq \epsilon F_{S_j}(e)\right] \leq 2\exp\left(-\frac{\epsilon^2\alpha F_{S_j}(e)}{3F(e)}\right) \leq 2\exp\left(-\frac{\epsilon^2\alpha(1-c)}{3}\right) \leq 1/n^3$$

Where the second inequality is due to the fact that $F_{S_j}(e)/F(e) \geq \min_{S \subseteq E, e' \in E \setminus S} F_S(e')/F(e') = 1 - c$, and the last transition is by setting an appropriate constant in $\alpha = \Theta(\frac{\log(n)}{\epsilon^2(1-c)})$. When the curvature is not bounded, we break the analysis into cases.

$F(e) \leq \gamma F_{S_j}(e)$**:** Setting $\mu = F_{S_j}(e)$ we get:

$$\mathbb{P}\left[|\hat{F}^j_{S_j}(e) - F_{S_j}(e)| \geq \epsilon F_{S_j}(e)\right] \leq 2\exp\left(-\frac{\epsilon^2\alpha F_{S_j}(e)}{3F(e)}\right) \leq 2\exp\left(-\frac{\epsilon^2\alpha}{3\gamma}\right) \leq 1/n^3$$

$F(e) > \gamma F_{S_j}(e)$:     Here we can set $\mu = F(e)/\gamma \geq F_{S_j}(e)$ and get:

$$\mathbb{P}\left[|\hat{F}_{S_j}^j(e) - F_{S_j}(e)| \geq \epsilon F(e)/\gamma\right] \leq 2\exp\left(-\frac{\epsilon^2 \alpha F(e)}{3\gamma F(e)}\right) \leq 2\exp\left(-\frac{\epsilon^2 \alpha}{3\gamma}\right) \leq 1/n^3$$

Where in both cases the last inequality is by setting an appropriate constant in $\alpha = \Theta(\epsilon^{-2}\gamma \log n)$. Note that in the first case we get an $(1 - \epsilon)$-approximate incremental oracle and in the second an additive $(\epsilon/\gamma)$-approximate incremental oracle. The second case is the worse of the two for our analysis. For both bounded and unbounded curvature, we take a union bound over all $e \in E$ and $j \in [k]$ (at most $n^2$ values), which concludes the proof. □

Note that in the above we use the fact that $F(e) \leq F(S^*)$ (recall that $S^*$ is the optimal solution for $F$) to get the second result. This is sufficient for our proofs to go through, however, the theorem has a much stronger guarantee which might be useful in other contexts (as we will see in Section 4).

## 3 EXPERIMENTS

We perform experiments on the following datasets, where the goal for all datasets is to maximize $F(S)$ subject to $|S| \leq k$.

**Uber pickups**     This dataset consists of Uber pickups in New York city in May 2014[5]. The set contains $652,434$ records, where each record consists of a longitude and latitude, representing a pickup location. Following (Rafiey & Yoshida, 2022) we aim to find $k$ positions for idle drivers to wait from a subset of popular pickup locations. We formalize the problem as follows. We run Lloyd's algorithm on the dataset, $X$, and find 100 cluster centers. We set these cluster centers to be the ground set $E$. We define the goal function $F : 2^E \to \mathbb{R}^+$ as $F(S) = \sum_{v \in X} f_v(S)$ where $f_v(S) = \max_{e \in E} d(v, e) - \min_{e \in S} d(v, e)$, and $d(v, e)$ is the Manhattan distance between $v$ and $e$.

**Discogs (Kunegis, 2013)**     This dataset[6] provides audio record information structured as a bipartite graph $G = (L, R; E')$. The left nodes represent labels and the right nodes represent styles. Each edge $(u, v) \in L \times R$ signifies the involvement of a label $u$ in producing a release of a style $v$. The dataset comprises $|R| = 383$ labels, $|L| = 243,764$ styles, and $|E'| = 5,255,950$ edges. We aim to select $k$ styles that cover the activity of the maximum number of labels. We construct a maximum coverage function $F : 2^R \to \mathbb{R}$, where $F(S) = \sum_{v \in L} f_v(S)$, and $f_v(S)$ equals 1 if $v$ is adjacent to some element of $S$ and 0 otherwise.

**Examplar-based clustering**     We consider the problem of selecting representative subset of $k$ images from a massive data set. We present experiments for both the CIFAR100 and the FashionMNIST datasets. For each dataset we consider a subset of $50,000$ images, denoted by $X$. We flatten every image into a one dimensional vector, subtract from it the mean of all images and normalize it to unit norm. We take the distance between two elements in $X$ as $d(x, x') = \|x - x'\|^2$. Here the ground set is simply the dataset, $E = X$. Similarly to the Uber pickup dataset we define the goal function $F : 2^E \to \mathbb{R}^+$ as $F(S) = \sum_{v \in X} f_v(S)$ where $f_v(S) = \max_{e \in E} d(v, e) - \min_{e \in S} d(v, e)$.

**Experimental setup**     We compare the sparsifier approach with the mini-batch approach for each of the above datasets as follows. We first compute the $p_i$'s and fix a parameter $\beta \in (0, 1)$. We take $\alpha$ in the algorithm such that $\alpha \sum_i^N p_i = \beta N$. That is, $\beta$ is the desired fraction of *data* (in our experiments the elements that make up $F$ correspondent to elements in the dataset) we wish to sample for the sparsifier / mini-batch. Note that the $p_i$'s are the same for the sparsifier and the mini-batch approach, and the only difference is whether we sample one time in the beginning (sparsifier) or for every iteration of the greedy algorithm (mini-batch). For example, setting $\beta = 0.01$ means that the sparsifier will sample 1% of the data and execute the greedy algorithm, while the mini-batch algorithm will sample 1% of the data for every iteration of the greedy algorithm.

In practice the naive algorithm is usually augmented with the following heuristic called *lazy-greedy* (Minoux, 2005). The algorithm initially creates a max heap of $E$ ordered according to a key $\rho(e)$, where initially $\rho(e) = F(e)$. During the $j$-th iteration it repeatedly pops $e$ from the top of the heap,

---

[5]https://www.kaggle.com/fivethirtyeight/uber-pickups-in-new-york-city
[6]http://konect.cc/networks/discogs_lstyle/

updates its key $\rho(e) = F_{S_j}(e)$ and inserts it back to the heap. Due to submodularity, if $e$ remains at the top of the heap after the update we know it maximizes $F_{S_j}(e)$. While this does not change the asymptotic number of oracle queries, it often leads to significant improvement in practice.

We also consider *stochastic-greedy* (Buchbinder et al., 2015; Mirzasoleiman et al., 2015), which further reduces the number of oracle evaluations by greedily choosing the next element added to $S_j$ from a subset of $E \setminus S_j$ of size $\frac{n}{k} \log \frac{1}{\epsilon}$ sampled uniformly at random.

We apply both of the above to the sparsifier / mini-batch approach and compare them against lazy-greedy and stochastic-greedy, respectively. Finally, for every mini-batch / sparsifier variant we execute we also run a baseline where $\forall i, p_i = 1/N$.

In Figure 1 we plot results for different $\beta$ values for both the sparsifier and the mini-batch approach for different values of $k$. Every point on the graph is the average of 20 executions with the same $k, \beta$. For every dataset we present three plots: (a) the relative utility (value of $F$) of the mini-batch / sparsifier approach compared to lazy-greedy, (b) the relative number of oracle evaluations excluding the preprocessing step, and (c) relative number of oracle evaluations including the preprocessing step. To allow different $\beta$ values to fit on the same plot we use a logarithmic scale in (b). On the other hand, the preprocessing step dominates in (c), therefore, we use a linear scale. We use "u" / "w" prefixes for uniform / weighted sampling.

In Figure 2 (Appendix D) we compare the sparsifier and mini-batch algorithm (augmented with the sampling scheme of stochastic-greedy) to the stochastic-greedy algorithm. As the ground sets for both Uber pickup and Discogs are rather small, we only run experiments on the image datasets.

**Results** We observe that the mini-batch algorithm is superior to the sparsifier approach for small values of $\beta$ while using about the same number of queries. What is perhaps most surprising is that uniform sampling outperforms over weighted sampling. We provide a theoretical explanation for this in the next section.

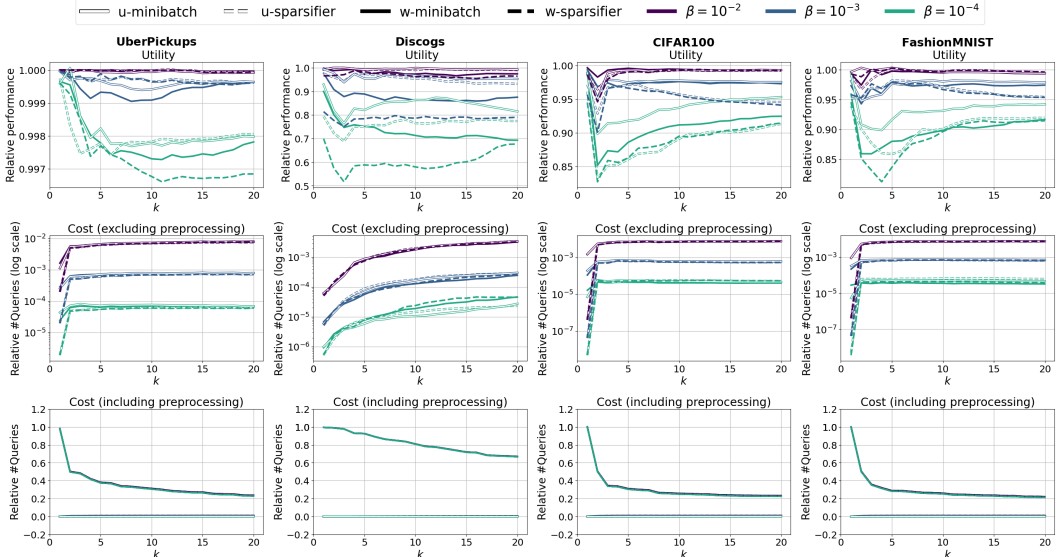

Figure 1: Sparsifier and mini-batch compared with lazy-greedy.

## 4 SMOOTHED ANALYSIS

In this section we show that uniform sampling achieves better results than weighted sampling under our models of smoothing. We start by noting that under uniform sampling it holds that $\sum_{i \in [N]} p_i = 1$ which in turn results in a bound of $\alpha$ in Lemma 2.1 instead of $n\alpha$ for the weighted case. This explains one part of the $\Theta(1/n\phi)$ factor in the transition from the "Weighted" to "Uniform" column in Table 1. For the rest of this section we explain the $1/\phi$ factor.

We start by stating the following useful Chernoff bound:

**Theorem 4.1** (Bounded dependency Chernoff bound (Pemmaraju, 2001))**.** *Let $X_1, ..., X_N$ be identically distributed random variables in the range $[0, 1]$ with bounded dependancy $d$. Let $X = \sum_{i=1}^{N} X_i$. Then for any $\epsilon \in [0, 1]$ and $\mu = \mathbb{E}[T]$ it holds that $\mathbb{P}(|X - \mathbb{E}[X]| \geq \epsilon\mu) \leq \Theta(d) \cdot \exp\left(-\Theta(\epsilon^2\mu/d)\right)$.*

**Model 1**    We show that Model 2 maintains the theoretical guarantees of both the sparsifier algorithm and the mini-batch algorithm under uniform sampling, with a multiplicative $\Theta(1/n\phi)$ factor in the query complexity.

**Theorem 4.2.** *Assuming uniform sampling ($p_i = 1/N$) and Model 1, the sparsifier construction of (Kenneth & Krauthgamer, 2023) and our mini-batch algorithm achieve the same guarantees, with an $\Theta(1/n\phi)$ multiplicative factor in the query complexity.*

*Proof.* Assumption (1) of Model 1 implies that $\mathbb{E}[F(e)] = \sum_{i \in [N]} \mathbb{E}[f^i(e)] \geq N\phi$. Applying the above Chernoff bound with $\epsilon = 1/2, \mu = \mathbb{E}[F(e)]$ we get that:

$$\mathbb{P}\left[|F(e) - \mathbb{E}[F(e)]| \geq \mu/2\right] \leq \Theta(d)\exp(-\Theta(\mu/d)) \leq \Theta(d)\exp(-\Theta(N\phi/d)) \leq 1/n^2$$

Where the last inequality is due to the fact that $N = \Omega((d/\phi)\log(nd))$. Finally, applying a union bound over all $e \in E$ we get that w.h.p $\forall e \in E, F(e) \geq N\phi/2$.

The above implies that w.h.p $p_i = \max_e f^i(e)/F(e) \leq 2/\phi N$, which implies that uniform sampling approximates the weights of weighted sampling up to a $O(1/\phi)$-factor. It was already noted by Rafiey & Yoshida (2022) that when given a multiplicative approximation of $\{p_i\}$ the sparsifier construction goes through with a respective multiplicative increase in the size of the sparsifier. This observation also applies to the results of Kenneth & Krauthgamer (2023) and for our results (Lemma 2.3). $\square$

**Empirical validation**    We can empirically evaluate $\phi$ by computing $\max_e \frac{1}{N}\sum_i f^i(e)$. The empirical values are as follows: CIFAR100: 0.34, FashionMNIST: 0.31, Uber pickup: $2.36 \times 10^{-4}$, Discogs: $4.10 \times 10^{-6}$. While Model 1 manages to explain the experimental results for some of the datasets, it appears that the model assumptions are a bit too strong.

**Model 2**    We show that Model 2 maintains the theoretical guarantees of the mini-batch algorithm under uniform sampling, with a multiplicative $\Theta(1/n\phi)$ factor in the query complexity.

**Lemma 4.3.** *Under Model 2, item (2) of Theorem 1.4 still holds with a $\Theta(1/n\phi)$ multiplicative factor in the query complexity for the mini-batch algorithm with uniform sampling.*

*Proof.* Similar to Theorem 4.2 it holds w.h.p that $F(e) \geq N\phi/2$, but now this is only guaranteed for a single element $e^* \in E$. We can use this fact to lower bound the optimum solution w.h.p: $F(S^*) \geq F(e^*) \geq N\phi/2$.

Recall $a = \max\{w_i f_S^i(e)\}_{i \in J}$ from Lemma 2.3, where $w_i = 1/\alpha_i$ and $\alpha_i = \min\{1, \alpha p_i\}$. Under uniform sampling it holds that $p_i = 1/N$. We conclude that $a \leq N/\alpha \leq 2F(e^*)/\phi\alpha$. We get an analogue to Lemma 2.3:

$$\mathbb{P}\left[|\hat{F}_S(e) - F_S(e)| \geq \epsilon\mu\right] \leq 2\exp\left(-\frac{\epsilon^2\mu}{3a}\right) \leq 2\exp\left(-\frac{\epsilon^2\alpha\phi\mu}{6F(e^*)}\right)$$

Plugging this into the second case (non-bounded curvature) of Theorem 1.3 we get that the same result still holds with an additional multiplicative $1/\phi$ factor in $\alpha$. $\square$

**Empirical validation**    Under Model 2 we can empirically evaluate $\phi$ by computing $\min_e \frac{1}{N}\sum_i f^i(e)$. The empirical values are as follows: CIFAR100: 0.38, FashionMNIST: 0.35, Uber pickup: 0.61, Discogs: 0.13. We conclude that $\phi = \Theta(1)$ and indeed Model 2 is able to explain the empirical success of the uniform mini-batch algorithm on all datasets.

**Discussion**    We observe that Model 2 manages to explain why uniform sampling outperforms in our experimental results. The empirical $\phi$ values are constant with respect to $n$. This means that the query complexity is actually less by about a $\Theta(n)$ factor for the uniform sampling case. We conclude that, given its speed and simplicity, the uniform mini-batch algorithm should be the first choice when tackling massive real-world datasets.

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

## A  SPARSIFIER BACKGROUND AND RELATED WORK

Rafiey & Yoshida (2022) were the first to showe how to construct a *sparsifier* for $F$. That is, given a parameter $\epsilon > 0$ they show how to find a vector $w \in \mathbb{R}^N$ such that the number of non-zero elements in $w$ is small in expectation and the function $\hat{F} = \sum_{i=1}^{N} w_i f^i$ satisfies with high probability (w.h.p)[7] that $\forall S \subseteq E, (1 - \epsilon)F(S) \leq \hat{F}(S) \leq (1 + \epsilon)F(S)$.

---

[7]Probability at least $1 - 1/n^c$ for an arbitrary constant $c > 1$. The value of $c$ does affect the asymptotics of the results we state (including our own).

Specifically, every $f^i$ is sampled with probability $\alpha_i$ proportional to $p_i = \max_{S \subseteq E, F(S) \neq 0} \frac{f^i(S)}{F(S)}$. If it is sampled, it is included in the sparsifier with weight $1/\alpha_i$, which implies that $\mathbb{E}[w_i] = 1$. While calculating the $p_i$'s exactly requires exponential time, Rafiey & Yoshida (2022) make do with an approximation, which can be calculated using interior point methods (Bai et al., 2016).

Rafiey & Yoshida (2022) show that if all $f^i$'s are non-negative and monotone[8], the above sparsifier can be constructed by an algorithm that requires $poly(N)$ oracle evaluations and the sparsifier will have expected size $O(\epsilon^{-2} B n^{2.5} \log n)$, where $B = \max_{i \in [N]} B_i$ and $B_i$ is the number of extreme points in the base polyhedron of $f^i$. They extend their results to matroid constraints of rank $r$ and show that a sparsifier with expected size $O(\epsilon^{-2} B r n^{1.5} \log n)$ can be constructed.

For the specific case of a cardinality constraint $k$, this implies a sparsifier of expected size $O(\epsilon^{-2} B k n^{1.5} \log n)$ can be constructed using $poly(N)$ oracle evaluations. The sparsifier construction is treated as a *preprocessing step*, and therefore the actual execution of Greedy on the sparsifier requires only $O(\epsilon^{-2} B k^2 n^{2.5} \log n)$ oracle evaluations to get a $(1 - 1/e - \epsilon)$ approximation. This is an improvement over Greedy when $N \gg n, B$.

Recently, Kudla & Zivný (2023) showed improved results for the case of *bounded curvature*. The *curvature* of a submodular function $F$ is defined as $c = 1 - \min_{S \subseteq E, e \in E \setminus S} \frac{F_S(e)}{F(e)}$. We say that $F$ has *bounded-curvature* if $c < 1$. Submodular functions with bounded curvature (Conforti & Cornuéjols, 1984) offer a balance between modularity and submodularity, capturing the essence of diminishing returns without being too extreme.

They show that when the curvature of all $f^i$'s and of $F$ is constant it is possible to reduce the preprocessing time to $O(Nn)$ oracle queries and to reduce the size of the sparsifier by a factor of $\sqrt{n}$. Furthermore, their results extend to the much more general case of *k-submodular functions*. While this significantly improves over the number of oracle calls compared to (Rafiey & Yoshida, 2022), the runtime of the preprocessing step depends on $\log\left(\max_{i \in [N]} \frac{\max_{e \in E} f^i(e)}{\min_{e \in E, f^i(e) > 0} f^i(e)}\right)$.

Note that the $B$ factor in (Kudla & Zivný, 2023; Rafiey & Yoshida, 2022) can be exponential in $n$ in the worst-case.

The current state of the art is due to Kenneth & Krauthgamer (2023) where they show that by sampling according to $p_i = \max_{e \in E, F(e) \neq 0} \frac{f^i(e)}{F(e)}$ it is possible to get both a fast sparsifier construction time of $O(Nn)$ and a small sparsifier size of $O(\frac{n^3}{\epsilon^2})$. Their analysis also implies that if the solution size is bounded by $k$ (e.g., a cardinality constraint) a sparsifier of size $O(\frac{nk^2}{\epsilon^2})$ is sufficient. They also present results for general submodular functions, however we only focus on their results for monotone functions which are relevant for this paper.

## B $p$-SYSTEMS

$p$-**systems**   The concept of $p$-systems offers a generalized framework for understanding independence families, parameterized by an integer $p$. We can define a $p$-system in the context of an independence family $\mathcal{I} \subseteq 2^E$ and $E' \subseteq E$. Let $\mathcal{B}(E')$ be the maximal independent sets within $\mathcal{I}$ that are also subsets of $E'$. Formally, $\mathcal{B}(E') = \{A \in \mathcal{I} | A \subseteq E' \text{ and no } A' \in \mathcal{I} \text{ exists s.t } A \subset A' \subseteq E'\}$. A distinguishing characteristic of a $p$-system is that for every $E' \subseteq E$, the ratio of the sizes of the largest to the smallest sets in $\mathcal{B}(E')$ does not exceed $p$: $\frac{\max_{A \in \mathcal{B}(E')} |A|}{\min_{A \in \mathcal{B}(E')} |A|} \leq p$.

The significance of $p$-systems lies in their ability to encapsulate a variety of combinatorial structures. For instance, when the intersection of $p$ matroids can be described using $p$-systems. In graph theory, the collection of matchings in a standard graph can be viewed as a 2-system. Extending this to hypergraphs, where edges might have cardinalities up to $p$, the set of matchings therein can be viewed as a $p$-system.

---

[8]Rafiey & Yoshida (2022) also present results for non-monotone functions, however, Kudla & Zivný (2023) point out an error in their calculation and note that the results only hold when all $f^i$'s are monotone.

**The greedy algorithm for $p$-systems** Formally, the optimization problem can be expressed as: $\max_{S \in \mathcal{I}} F(S)$ where the pair $(E, \mathcal{I})$ characterizes a $p$-system and $F : 2^E \to \mathbb{R}^+$ denotes a non-negative monotone submodular set function. It was shown by Nemhauser et al. (1978) that the natural greedy approach achieves an optimal approximation ratio of $\frac{1}{p+1}$. Setting $A_j = \{e \mid S_j + e \in \mathcal{I}\}$ (i.e., $S_j$ remains an independent set after adding $e$) in Algorithm 1 we get the greedy algorithm of Nemhauser et al. (1978). Note that for general $p$-systems it might be that $k = n$, however, there are very natural problems where $k \ll n$. For example, for maximum matching $E$ corresponds to all edges in the graph, which can be quadratic in the number of nodes, while the solution is at most linear in the number of nodes.

## C  MISSING PROOFS

**Proof of Lemma 2.1** We start by showing that $\sum_{i=1}^{N} p_i \leq n$. Let us divide the range $[N]$ into

$$A_e = \left\{ i \in N \mid e = \underset{e' \in E, F(e') \neq 0}{\arg\max} \frac{f^i(e')}{F(e')} \right\}$$

If 2 elements in $E$ achieve the maximum value for some $i$, we assign it to a single $A_e$ arbitrarily.

$$\sum_{i=1}^{N} p_i = \sum_{i=1}^{N} \max_{e \in E} \frac{f^i(e)}{F(e)} = \sum_{e \in E} \sum_{i \in A_e} \frac{f^i(e)}{F(e)} = \sum_{e \in E} \frac{\sum_{i \in A_e} f^i(e)}{F(e)} \leq \sum_{e \in E} 1 \leq n$$

Let $X_i$ be an indicator variable for the event $w_i > 0$. We are interested in $\sum_{i=1}^{N} \mathbb{E}[X_i]$. It holds that:

$$\sum_{i=1}^{N} \mathbb{E}[X_i] = \sum_{i=1}^{N} \alpha_i \leq \sum_{i=1}^{N} \alpha p_i = \alpha \sum_{i=1}^{N} p_i \leq \alpha n$$

**Proof of Theorem 1.2** Theorem 1.2 directly follows from the two lemmas below.

**Lemma C.1.** *Let $\epsilon' \leq \epsilon/2k$. Algorithm 1 with an additive $\epsilon'$-approximate incremental oracle achieves a $(1 - 1/e - \epsilon)$-approximation under a cardinality constraint $k$.*

*Proof.* Let $S^*$ be some optimal solution for $F$. We start by proving that the following holds for every $j \in [k]$:

$$F(S_{j+1}) - F(S_j) \geq \frac{1}{k}((1 - \epsilon)F(S^*) - F(S_j))$$

Fix some $j \in [k]$ and let $S^* \setminus S_j = \{e_1^*, \ldots, e_\ell^*\}$ where $\ell \leq k$. Let $S_t^* = \{e_1^*, \ldots, e_t^*\}$, and $S_0^* = \emptyset$. Let us first use submodularity and monotonicity to upper bound $F(S^*)$.

$$F(S^*) \leq F(S^* + S_j)$$

$$= F(S_j) + \sum_{t=1}^{\ell} [F(S_j + S_t^*) - F(S_j + S_{t-1}^*)]$$

$$\leq F(S_j) + \sum_{t=1}^{\ell} F_{S_j}(e_t^*) \leq F(S_j) + \sum_{t=1}^{\ell} \max_{e \in E \setminus S_j} F_{S_j}(e)$$

$$\leq F(S_j) + k \max_{e \in E \setminus S_j} F_{S_j}(e)$$

$$\leq F(S_j) + k(\max_{e \in E \setminus S_j} \hat{F}_{S_j}^j(e) + \epsilon' F(S^*))$$

Where the last inequality is due to the fact that $\hat{F}_{S_j}^j$ is an additive $\epsilon'$-approximate incremental oracle.

Noting that $e_j = \arg\max_{e \in E \setminus S_j} \hat{F}_{S_j}^j(e)$ we get that:

$$F(S^*) \leq F(S_j) + k(\hat{F}_{S_j}^j(e_j) + \epsilon' F(S^*))$$

$$\implies \hat{F}_{S_j}^j(e_j) \geq \frac{1}{k}((1 - \epsilon' k)F(S^*) - F(S_j))$$

The above lower bounds the progress on the $j$-th mini-batch. Now, let us bound the progress on $F$. Again, we use the fact that $\hat{F}_{S_j}^j$ is an additive $\epsilon'$-approximate incremental oracle.

$$F(S_{j+1}) - F(S_j) \geq \hat{F}_{S_j}^j(e_j) - \epsilon' F(S^*)$$

$$\geq \frac{1}{k}((1 - \epsilon' k)F(S^*) - F(S_j)) - \epsilon' F(S^*)$$

$$\geq \frac{1}{k}((1 - 2\epsilon' k)F(S^*) - F(S_j))$$

Finally, using the fact that $\epsilon' \leq \epsilon/2k$ we get:

$$F(S_{j+1}) - F(S_j) \geq \frac{1}{k}((1 - \epsilon)F(S^*) - F(S_j))$$

Rearranging, the result directly follows using standard arguments.

$$F(S_{k+1}) > \frac{(1 - \epsilon)}{k}F(S^*) + (1 - \frac{1}{k})F(S_k)$$

$$\geq \frac{(1 - \epsilon)}{k}F(S^*)(\sum_{i=0}^{k}(1 - \frac{1}{k})^i) + F(\emptyset)$$

$$\geq F(S^*)\frac{(1 - \epsilon)(1 - \frac{1}{k})^k}{k(1 - (1 - \frac{1}{k}))} = (1 - \epsilon)(1 - \frac{1}{k})^k F(S^*)$$

$$\geq (1 - \epsilon)(1 - 1/e)F(S^*) \geq (1 - 1/e - \epsilon)F(S^*)$$

$\square$

**Lemma C.2.** *Let $\epsilon' \leq \epsilon/2kp$. Algorithm 1 with an additive $\epsilon'$-approximate incremental oracle achieves a $(\frac{1-\epsilon}{1+p})$-approximation under a $p$-system constraint.*

*Proof.* Let $S^*$ be some optimal solution for $F$. Assume without loss of generality that the solution returned by the algorithm consists of $k$ elements $S_{k+1} = \{e_1, \dots, e_k\}$.

We show the existence of a partition $S_1^*, S_2^*, \dots, S_k^*$ of $S^*$ such that $F_{S_j}(e_j) \geq \frac{1}{p}F_{S_{k+1}}(S_j^*) - 2\epsilon' F(S^*)$. Note, we allow some of the sets in the partition to be empty.

Define $T_k = S^*$. For $j = k, k - 1, \dots, 2$ execute: Let $B_j = \{e \in T_j \mid S_j + e \in \mathcal{I}\}$. If $|B_j| \leq p$ set $S_j^* = B_j$; else pick an arbitrary $S_j^* \subset B_j$ with $|S_j^*| = p$. Then set $T_{j-1} = T_j \setminus S_j^*$ before decreasing $j$. After the loop set $S_1^* = T_1$. It is clear that for $j = 2, \dots, k, |S_j^*| \leq p$.

We prove by induction over $j = 0, 1, \dots, k - 1$ that $|T_{k-j}| \leq (k - j)p$. For $j = 0$, when the greedy algorithm stops, $S_{k+1}$ is a maximal independent set contained in $E$, therefore any independent set (including $T_k = S^*$) satisfies $|T_k| \leq p|S_{k+1}| = pk$. We proceed to the inductive step for $j > 0$. There are two cases: (1) $|B_{k-j+1}| > p$, which implies that $|S_{k-j+1}^*| = p$ and using the induction hypothesis we get that $|T_{k-j}| = |T_{k-j+1}| - |S_{k-j+1}^*| \leq (k - j + 1)p - p = (k - j)p$. (2) $|B_{k-j+1}| \leq p$, it holds that $T_{k-j} = T_{k-j+1} \setminus B_{k-j+1}$. Let $Y = S_{k-j+1} + T_{k-j}$. Due to the definition of $B_{k-j+1}$ it holds that $S_{k-j+1}$ is a maximal independent set in $Y$. It holds that $T_{k-j}$ is independent and contained in $Y$, therefore $|T_{k-j}| \leq p|S_{k-j+1}| = p(k - j)$.

Finally, we get that $|T_1| = |S_1^*| \leq p$. By construction it holds that $\forall j \in [k], \forall e \in S_j^*, S_j + e$ is independent. From the choice made by the greedy algorithm and the fact that $\hat{F}_{S_j}^j$ is an additive $\epsilon'$-approximate incremental oracle it follows that for each $e \in S_j^*$:

$$F_{S_j}(e_j) \geq \hat{F}_{S_j}^j(e_j) - \epsilon' F(S^*)$$

$$\geq \hat{F}_{S_j}^j(e) - \epsilon' F(S^*) \geq F_{S_j}(e) - 2\epsilon' F(S^*)$$

Hence,

$$\left|S_j^*\right| F_{S_j}(e_j) \geq \sum_{e \in S_j^*} \left(F_{S_j}(e) - 2\epsilon' F(S^*)\right)$$

$$\geq F_{S_j}(S_j^*) - 2\epsilon' \left|S_j^*\right| F(S^*) \geq F_{S_{k+1}}(S_j^*) - 2\epsilon' \left|S_j^*\right| F(S^*)$$

Using submodularity in the last two inequalities.

For all $j \in \{1, 2, ..., k\}$ it holds that $\left|S_j^*\right| \leq p$, and thus $F_{S_j}(e_j) \geq \frac{1}{p} F_{S_{k+1}}(S_j^*) - 2\epsilon' F(S^*)$. Using the partition we get that:

$$F(S_{k+1}) \geq \sum_{j=1}^{k} F_{S_j}(e_j) \geq \sum_{j=1}^{k} \left(\frac{1}{p} F_{S_{k+1}}(S_j^*) - 2\epsilon' F(S^*)\right)$$

$$\geq \frac{1}{p} F_{S_{k+1}}(S^*) - 2\epsilon' k F(S^*)$$

$$\geq \frac{1}{p} (F(S^*) - F(S_{k+1})) - 2\epsilon' k F(S^*)$$

Where the second to last inequality is due to submodularity and the last is due to monotonicity. Rearranging we get that:

$$F(S_{k+1}) \geq \frac{(1 - 2p\epsilon' k)}{p+1} F(S^*)$$

As $\epsilon' < \frac{\epsilon}{2pk}$ we get the desired result. $\qquad\square$

## D  ADDITIONAL EXPERIMENTS

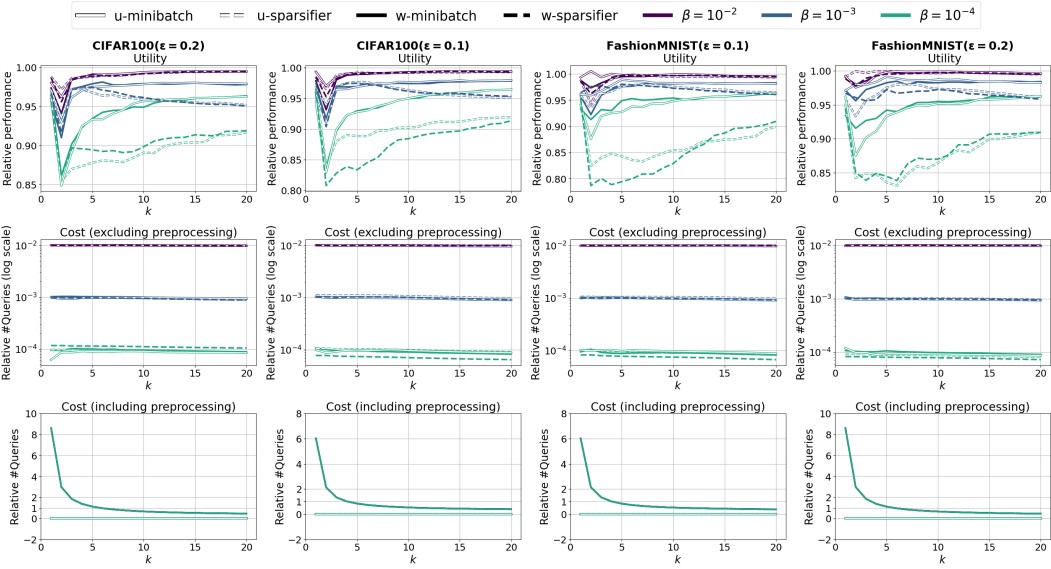

Figure 2: Sparsifier and mini-batch compared with stochastic-greedy for $\epsilon = 0.1$ and $\epsilon = 0.2$.

