# OpenReview forum: "Mini-batch Submodular Maximization"
_ICLR.cc/2025/Conference — Submitted to ICLR 2025_

### Official Review · Reviewer_3AMz · 2024-11-03

**Soundness:** 3
**Presentation:** 3
**Contribution:** 2
**Rating:** 5
**Confidence:** 4

**Summary:**

This work studies a sampling-based algorithm for faster non-negative monotone *decomposable* submodular maximization subject to
cardinality or $p$-system constraints. In particular, it builds on work of
[Kenneth-Krauthgamer, ICALP 2024] (please update reference in paper), which sparsifies
and reweights the set of functions $f^{(i)}(S)$ for the input function $F(S) = \sum_{i=1}^N f^{(i)}(S)$.
The goal of this paper is to eliminate the dependence on $N$, which the authors do under mild assumptions
via *smoothed analysis*. They also show that this is not possible in the general case with a simple pathological example.
In short, the main idea is to sample a subset of $f^{(i)}(S)$ functions at each step to form a
"mini-batch" for approximating the full $F(S)$. The algorithm then greedily
select the next element based on the sampled funciton (which changes in each iteration), not $F$ itself.

Further, under the mild realistic assumptions, they prove why uniform sampling is a competitive approach,
which helps explain initially surprising experimental observations.
Lastly, this work provides a clean set of experiments comparing their mini-batch sampling-based methods to
a full lazy greedy algorithm and the sparsification idea in [Kenneth-Krauthgamer, ICALP 2024].

**Strengths:**

- Uses smoothed analysis to more accurately study realistic inputs
- Table 1 cleanyl describes the results, including a comparison with [Kenneth-Krauthgamer, ICALP 2024]
- Draws connections to the lazier-than-lazy greedy algorithm of [Mirzasoleiman et al., AAAI 2015]
  and explains how the two ideas can be combined to reduce query complexity by a factor of $\Theta(k)$
- Good comprehensive set of experiments for cardinality constraints, though the
  values of $k \le 20$ are quite small. It would be nicer to increase $k$ to see
  how fast the different algorithms converge (relatively) to lazy greedy

**Weaknesses:**

- The lunch menu optimization example, while a clear illustration, does not
  really motivate the problem from a practioner's perspective
- There are no $p$-system experiments
- It is unclear if ICLR is an appropriate venue for this work. The
  non-exhaustive list of topics in the Call for Papers includes "optimization",
  but submodular maximization in its raw form seems one hop away from the
  target areas of ICLR (deep learning)

**Questions:**

- In the introduction, you claim that "in many of the above applications, $N$
  (the number of underlying submodular functions) is extremely large, making the
  evaluation of $F$ prohibitively slow." Are there realistic examples where $N
  \gg 1000$? It's not clear to me how often we really encounter $N$ *distinct*
  personalized submodular functions.
- What exactly is the quantity $A_e$ when you first introduce it on page 3?
  This should be made more clear. Initially, I thought it was a vector of all
  marginal values, but then in model 1 you say it's a random variable.
- For the Uber pickups experiment, why do you use Llyod's algorithm to find
  centers instead of a data-indepedndent grid?

---

> ### Author Response · Authors · 2024-11-13
>
> Thank you for the review. We address your comments below.
>
> Weaknesses:
> - "The lunch menu optimization example, while a clear illustration, does not really motivate the problem from a practioner's perspective"
>
> Agreed, we chose this for simplicity. However, utility maximization is an extremely natural problem. Other examples for utility maximization include: adding medical services to a healthcare package as to maximize the welfare of all patients, adding features to a website to maximize user engagement, etc...
>
> - "There are no p-system experiments"
>
> Indeed, we couldn't find a real-world dataset for this problem. Previous papers seems to either be completely theoretical (no experiments), or run experiments just under a cardinality constraint.
>
> Questions:
>
> Q1) It is quite natural in welfare maximizations (e.g., many people with different preferences). Another example is finding a representative set of images (e.g., thumbnails for a video). Here N can be very large (the number of frames in the video), clearly there is plenty of redundancy, so our approach is very natural here.
>
> Q2) It is defined just above Model 1. It is the set $\{f^i(e)\}_{i\in [N]}$ and in our models we assume that every $f^i(e)$ (the value of the i-th func on e) is a random variable, not the set $A_e$.
>
> Q3) We roughly followed the paper of Rafiey and Yoshida which introduced this set. They simply say that they select a set of "popular pickup locations in the dataset". We used k-means to pick "popular locations".

---

> > ### Comment · Reviewer_3AMz · 2024-11-18
> >
> > Thank you for your response. I have read through all the reviews and rebuttals, and will maintain my score.

---

> > > ### Author Response · Authors · 2024-11-23
> > >
> > > Thank you for your time. In order to improve the paper for future submissions, may we ask what improvement in your opinion would make the paper cross the acceptance threshold?

---

> ### Author Response · Authors · 2024-11-15
>
> Another point we would like to address:
> "It is unclear if ICLR is an appropriate venue for this work. The non-exhaustive list of topics in the Call for Papers includes "optimization", but submodular maximization in its raw form seems one hop away from the target areas of ICLR (deep learning)"
>
> Submodular optimization papers are often published in ICLR / Neurips / ICML. While it is true that usually there are only a few submissions dealing with the raw form of submodular optimization, they are accepted quite positively. See for example this submission for ICLR 2025 which received some very positive reviews - https://openreview.net/forum?id=EPHsIa0Ytg

---

### Official Review · Reviewer_87eG · 2024-11-04

**Soundness:** 3
**Presentation:** 4
**Contribution:** 2
**Rating:** 6
**Confidence:** 4

**Summary:**

This paper addresses the problem of maximizing a non-negative, monotone, decomposable submodular function under the cardinality constraint and $p$-system constraint. It introduces the first mini-batch algorithm with weighted sampling for this problem, demonstrating that it outperforms the sparsifier-based approach both theoretically and empirically. Additionally, the authors observe that, in experiments, uniform sampling outperforms weighted sampling. To explain this outcome, they define two smoothing models. The first model provides theoretical guarantees for both the mini-batch and sparsifier algorithms on some datasets, while the second model applies only to the mini-batch algorithm but is effective across all datasets tested.

**Strengths:**

Overall, the paper is well-structured and easy to understand. The definitions and explanations are clear, and related work is discussed in sufficient detail.

The discussion on uniform and weighted sampling, along with the smoothing model, helps bridge the gap between theoretical results and the empirical performance of the algorithms. It provides insights into why an algorithm without a worst-case guarantee can still perform well in experiments.

**Weaknesses:**

The algorithm is simple, and the analysis is quite straightforward. The technical contribution is limited.

With 12 indistinguishable lines in Figure 1, it is hard to see which algorithm with $\beta=10^{-2}$ achieves the best performance.

**Questions:**

It might be better to put Section 4 before Section 3 to ensure the continuity of the analysis.

---

> ### Author Response · Authors · 2024-11-13
>
> Thank you for your review.
>
> About the figure, basically all algorithms achieve almost the same quality when we sample enough elements.
> We will move Section 4 before Section 3 if accepted.

---

### Official Review · Reviewer_ARiB · 2024-11-04

**Soundness:** 3
**Presentation:** 3
**Contribution:** 1
**Rating:** 3
**Confidence:** 4

**Summary:**

This paper considers maximization of decomposable monotone submodular functions over a ground set of size $n$, meaning that the objective function $f$ is a sum of $N$ monotone submodular functions $f_1,...,f_N$. If $N$ is large, then evaluations of $f$ may be computationally demanding. Previous work on the topic (Rafiey & Yoshida, 2022; Kenneth & Krauthgamer 2023) proposes constructing a random sparsified version of $f$ that is a weighted sum of some subset of the functions, and is within a multiplicative $\epsilon$ factor approximation on all sets. A sparsifier such as those mentioned could be constructed as a preprocessing step for an algorithm, and then the algorithm would be run using the sparsifier in place of the original function. The state of the art is that of Kenneth & Krauthgamer, where a sparsifier of $O(k^2n\epsilon^{-2})$ functions is constructed using $O(Nn)$ oracle calls. The sparsifier is constructed by iterating over the functions, computing a probability $p_i$ for each function $f_i$ to be included, and then sampling that function with probability $p_i$ (which takes a total of $O(Nn)$ queries). Then querying the sparsifier takes $O(k^2n\epsilon^{-2})$ function evaluations, compared to $O(N)$ function evaluations to query the original $f$. If $N$ is relatively large, the sparsifier is more efficient.

Instead of computing a sparsifier as a preprocessing step for an algorithm, this paper proposes a "mini-batch method" (which have been used in other areas of ML) for this problem (Algorithm 3). That is, a new sparsifier is sampled every iteration of the greedy algorithm. The approach in this paper uses the same sampling probabilities $p_i$ as Kenneth & Krauthgamer, and therefore still needs the $O(Nn)$ queries as a preprocessing step to compute the $p_i$. In order to prove some of the results in their paper, they make additional assumptions on the problem setting (Models 1 and 2). Several analyses are done on the number of function queries needed for their algorithm. Finally, they include an experimental comparison of their algorithm and related works.

**Strengths:**

- Exploring submodular optimization algorithms that do not view the function $f$ as simply a black box is an interesting research direction that I think deserves attention.
- They explained their results clearly and the paper was easy to understand.

**Weaknesses:**

- It seems a lot of the difficulty of these sparsification approaches is because the sampling of the $f_i$ is non-uniform, but it is still unclear to me that this is so much better than uniform sampling. According to this paper, uniform sampling does better in practice, and requires no preprocessing to compute the $p_i$ since they would be uniform. It is also stated that no theoretical bound can be gotten for uniform sampling. But if we assume that all the $f_i$ are bounded by some value $R$, why can't concentration inequalities be used to get a theoretical guarantee for the uniform approach?
- Some of the results are dependent on assuming Models 1 or 2 (see Table 1), but it isn't clear to me that these models are realistic for applications of the problem.
- Improvements over Kenneth and Krauthgamer mainly include the curvature of the function in the bound on the number of function queries, so the bounds are instance dependent.
- The bounded curvature results (which don't depend on Models 1 and 2) don't use ideas that are that novel compared to related work. It seems the biggest difference from Kenneth and Krauthgamer is computing the sparsifier at each round of the greedy algorithm, and only relatively minor changes are needed to the argument of Kenneth and Krauthgamer.

**Questions:**

* If the $f_i$ are all bounded by a value $R$, could theoretical guarantees be gotten for uniform sampling?
* Do you expect Models 1 and 2 would hold widely in applications of decomposable functions?

---

> ### Author Response · Authors · 2024-11-13
> **Our main contribution is the uniform alg + smoothed analysis.**
>
> Thank you for your review.
>
> We would like to emphasize that our main contribution is the introduction of the *uniform* sampling algorithm, observing that it outperforms other approaches empirically, and using smoothed analysis to bridge the gap between theory (no worst case analysis possible) and practice. We believe that this algorithm can be used as the first line of attack for many real-world massive datasets. The improved weighted sampling is "nice to have" and lays the groundwork for the smoothed analysis of the uniform sampling algorithm, but this is not our main contribution.
>
> We address your questions below.
>
> Q1) An upper bound is not sufficient. This is because our proofs require a *multiplicative* error bound for the minibatch approach to work. Consider the following example: all functions except one are always zero ($f^i \equiv 0, i\neq j$), and one function, $f^j$, is upper bounded by 1. Clearly both the minibatch and the sparsifier algorithms can't optimize the sum as they will keep sampling functions that are always 0. The above example is unlikely to appear in real world applications, but it illustrates that worst-case analysis is simply not the right tool here. This is why we use smoothed analysis to explain the superior performance of uniform sampling in practice.
>
> Q2) Yes, specifically Model 2. The assumptions of Model 2 are *extremely* mild and we verify empirically that they hold for *all* of our datasets. We would like to emphasize that we only introduced smoothed analysis in this revision of the paper, while we used the same datasets in previous revisions. That is, we did not simply pick datasets where our models apply (and indeed Model 1 does not apply to all datasets).

---

> ### Author Response · Authors · 2024-11-23
>
> Dear reviewer, the discussion period is ending soon, and we would really appreciate your response.

---

### Author Response · Authors · 2024-12-04
**Summary**

Unfortunately, there was very limited interaction during the discussion period. Due to the public nature of the discussion, I'm adding this summary to clarify key points and avoid misunderstandings.

The reviews mostly lacked constructive feedback. The main critique focused on the simplicity of our analysis, which is actually a strength, and overlooked the core contribution: our uniform sampling mini-batch algorithm. This approach outperforms over other approaches both theoretically and practically. It is easy to implement and can be used for massive datasets.

When considering potential impact, a good reference is Sculley's mini-batch k-means algorithm (https://dl.acm.org/doi/10.1145/1772690.1772862), which is extremely simple (just 2 pages) yet widely used in practice (e.g., sklearn).

---

### Meta-Review · Area_Chair_i4Qy · 2024-12-21

**Metareview:**

The paper studies the problem of maximizing a decomposable submodular function subject to constraints. The main contribution of the paper is a sampling-based algorithm that uniformly samples and reweighs a subset of the functions in order to create a much smaller sparsified instance that can be solved more efficiently. Under certain assumptions , the paper uses smoothed analysis to show that the uniform sampling approach improves upon existing sparsifier approaches.

One of the main strengths of this work is that it provides a theoretical justification for the uniform sampling approach which is a preferred approach in practice. Although this work makes a valuable contribution that is relevant to applications, there was consensus among the reviewers that the contribution is limited and it does not meet the threshold for acceptance.

**Additional Comments On Reviewer Discussion:**

During the discussion, the authors clarified several potential misunderstandings regarding the algorithm and its analysis. The authors also addressed the reviewers' questions. The main concern raised was that the contribution is limited, which is the main factor in the decision.

---

### Decision · Program_Chairs · 2025-01-22

Reject